# Physiochemical Characterization of Biochars from Six Feedstocks and Their Effects on the Sorption of Atrazine in an Organic Soil

Shagufta Gaffar [1], Sanku Dattamudi [1], Amin Rabiei Baboukani [2], Saoli Chanda [1], Jeffrey M. Novak [3], Donald W. Watts [3], Chunlei Wang [2] and Krishnaswamy Jayachandran [1,*]

[1] Department of Earth and Environment, Institute of Environment, Florida International University, Miami, FL 33199, USA; sgaff008@fiu.edu (S.G.); sdattamu@fiu.edu (S.D.); schanda@fiu.edu (S.C.)
[2] Department of Mechanical and Materials Engineering, Florida International University, Miami, FL 33174, USA; Arabi009@fiu.edu (A.R.B.); wangc@fiu.edu (C.W.)
[3] USDA-ARS-Coastal Palin Soil, Water and Plant Research Center, 2611 W Lucas St, Florence, SC 29501, USA; jeff.novak@usda.gov (J.M.N.); don.watts@ars.usda.gov (D.W.W.)
* Correspondence: jayachan@fiu.edu; Tel.: +1-305-348-6553

**Abstract:** Application of biochars in agricultural soils has the potential to reduce groundwater contamination of atrazine, a widely used herbicide in the US, therefore sustaining environmental quality and reducing human health issues. This study was conducted to characterize biochars produced from six feedstocks and investigate their ability to remove and retain atrazine in an organic-rich soil. Australian pine (AP), Brazilian pepper (BP), coconut husk (CH), cypress (Cy), loblolly pine (L), and pecan shell (P) feedstocks were pyrolyzed at 350 °C and 500 °C. Adsorption and desorption behaviors of atrazine were explained using Freundlich isotherms. Higher pyrolysis temperature increased specific surface area (5 times), total pore volume (2.5 times), and aromaticity (1.4 times) of the biochars. CH feedstock produced the most effective biochars (CH350 and CH500), which adsorb 8–12% more atrazine than unamended soils. CH350 biochar performed the best ($K_{d\ ads}$ = 13.80, $K_{OC}$ = 153.63, $K_{d\ des}$ = 16.98) and had significantly higher ($p < 0.05$) adsorption than unamended soil, possibly resulting from its highest cation exchange capacity (16.32 cmol kg$^{-1}$). The $K_{d\ des}$ values for atrazine desorption were greater than the $K_{d\ ads}$ for adsorption, indicating retention of a considerable amount of atrazine by the biochar-amended soils following desorption.

**Keywords:** biochar; adsorption; desorption; atrazine; hysteresis

## 1. Introduction

Atrazine (2-chloro-4-ethylamino-6-isopropylamino-1,3,5-trithemazine) is a very common herbicide (triazine group) used in the United States to control pre- and post-emergence broadleaf weeds (*Stellaria media*, *Taraxacum officinale*, *Lespedeza cuneata*, etc.) in agricultural production. Herbicide accounted for approximately 59% of the total pesticide used in the US agricultural sector and about 29.03 to 33.57 million kg of atrazine was used for agricultural purposes in the US in 2012 [1]. Atrazine application was reported to save about USD 2.9 billion every year in corn production in the US [2]. However, atrazine can remain in the soil for several days to months when applied and is often decomposed relatively quickly by soil microorganisms (such as *Arthrobacter*, *Nocardioides*) which, as a result, easily contaminate ground and surface water systems. In recent decades, atrazine and its metabolites, namely desethyl-atrazine (DEA; 2-amino-4-chloro-6-(isopropylamino)-*s*-triazine), deisopropyl-atrazine (DIA; 2-amino-4 chloro-6-(ethylamino)-*s*-triazine), and hydroxyl-atrazine (HA; 2-hydroxy-4-(ethylamino)-6-(isopropylamino)-*s*-triazine) have been commonly detected in soil, drinking water aquifers, shallow groundwater and in surface water ([3]. This is an imperative area of concern because atrazine has been recognized as an endocrine disruptor compound in humans [4]. It also has adverse effects on the immune

and central nervous systems of other mammals and aquatic invertebrates [5]. Therefore, an important issue remains to control the availability of atrazine in soil solution and its fate and transport in natural water resources.

Sorption, a common physiochemical process, is an effective solution for controlling the loss of hydrophobic organic compounds (HOCs) like atrazine in the environment. Biochar, a byproduct of thermal pyrolysis of carbon-rich biomass, is often used as a soil amendment in agricultural fields [6] and has the potentiality to adsorb HOCs when applied and thus reduce their loss from the soil profile. Biochar is also an effective agent for accumulating soil organic C (SOC), reducing greenhouse gas emissions from the field [7,8], increasing soil nutrient availability [9,10] and influencing soil microbial activity [11]. However, feedstock types, pyrolysis temperature, method of pyrolysis and other factors can influence biochar properties and their effects on crop production and environmental sustainability [12]. Therefore, the characterization of biochar is an important initial step in understanding the production specifics and application mechanisms in soil systems.

Adsorption is usually the first process that begins straightaway when pesticides are applied to soil. One of the important factors influencing the effectiveness of biochar for pesticide retention is pyrolysis condition [13]. Increasing pyrolysis temperature can increase surface area, C content and aromaticity of the biochars along with the decrease in polarity and oxygen and hydrogen contents [14,15]. Consequently, the potential of biochar to sorb organic contaminants makes it a unique adsorbent [15,16]. However, biochar produced at low pyrolysis temperatures (250 to 400 °C) is characterized by higher polarity and the amount of oxygen-containing functional groups on its surfaces, and consequently, is an effective agent in removing inorganic/polar organic contaminants [15,17].

Biochar can be made from almost any organic feedstock, however, the use of materials including agricultural wastes, forestry residues, dead biomass, urban yard waste, municipal solid wastes, etc. that do not compete with agricultural food production and would otherwise decompose must be taken into consideration as potential feedstocks for the sustainable production of biochar [18]. Invasive plant species are often considered as a great threat to the agricultural ecosystem by competing with native species for resources and even altering the chemical properties of the soil. A recent report indicated that the US alone spends about USD 3 billion to prevent, control and eradicate invasive plant species [19]. A major problem in dealing with invasive species, in addition to the cost involved in their management, has to do with their removal, extraction and the sustainable management of the waste (residual) products. Application of biochar made from invasive plant species can be an incentive to deal with these obnoxious plants (such as *Casaurina equisetifolia*, *Schinus terebinthifolius*, *Melaleuca quinquenervia*, etc.) in a profitable way. Moreover, the high temperature of the pyrolysis process can sterilize the invasive plant species which can contribute to preventing further spread of these biological pollutants. The effect of biochar on atrazine adsorption in mineral soil is a well-researched topic [20–22], however, the comparative analysis of different biochar feedstocks on sorption isotherms of atrazine in organic-rich soil is limited. We expect that the effect of biochars on organic-rich soil would not be the same as in the mineral soil. Therefore, this study is a unique approach to investigate sorption kinetics of atrazine in biochar-amended and unamended organic-rich soil for soil health and environmental quality assessment.

It was also recognized that a systematic study on the physical and chemical characterization of biochars made from native and invasive plant species and their performance appraisal on atrazine sorption behavior has not been adequately carried out before. Therefore, the objective of this study was to combine invasive plant species with native plants and agricultural residues as potential feedstocks for biochar production and characterize twelve different biochars made from a total of six different feedstocks at two different temperatures. The study specifically aimed to investigate the influence of the twelve different biochars on the adsorption and desorption of atrazine as an effective means for removing pesticides from the environment.

## 2. Materials and Methods

### 2.1. Soil Collection and Analysis

The soil (0–5 cm depth) used for this study was collected from the organic garden research plot (25.7540° N, 80.3801° W) at Florida International University (FIU). The organic garden is an outdoor research facility located at FIU consisting of plots to conduct experiments and is a fully organic system with no application of synthetic chemicals. The soil sample was air-dried, passed through a 2 mm sieve and homogenized prior to use. Soil textural class was analyzed by the hydrometer method using a Fisher brand ATSM 152H soil hydrometer. Soil pH was measured in a 1:2 (*w/v*) soil/deionized water mixture using a Denver instrument pH meter. The organic matter (OM) content was determined by the loss on ignition (LOI) method with a Fisher Scientific isotemp muffle furnace (at 550 °C). Selected physical and chemical soil properties are presented in Table 1.

**Table 1.** Physical and chemical properties of soils used for this experiment.

| Parameters | Mean Values |
| :---: | :---: |
| pH [†] | 7.52 ± 0.04 |
| C (%) [‡] | 9.90 ± 0.55 |
| N (%) [‡] | 0.55 ± 0.03 |
| OM (%) [§] | 15.49 ± 0.34 |
| OC (%) [§] | 8.98 ± 0.20 |
| Sand (%) [¶] | 76.44 |
| Silt (%) [¶] | 21.65 |
| Clay (%) [¶] | 1.91 |

[†] pH is measured in 1:2 soil and deionized water slurry (*w/v*); [‡] carbon and nitrogen were analyzed using a CN analyzer; [§] OM = organic matter content was calculated by loss on ignition method; [§] OC = organic carbon content was calculated from organic matter content; [¶] particle size distribution (sand, silt and clay) was calculated by hydrometer method; numbers are mean ± standard error.

### 2.2. Properties of Chemicals Used in This Study

An analytical grade of atrazine (2-chloro-4-ethylamino-6-isopropylamino-1,3,5-trithemazine) with ≥98% purity was purchased from Cayman Chemical, Michigan, USA. It has an aqueous solubility of 33 mg L$^{-1}$ at 22 °C and pH 7, a logP$_{ow}$ of 2.82 and a p*K*a value of 1.7 [23]. Deionized water was used to prepare a 20 mg L$^{-1}$ stock solution of atrazine. HPLC grade methanol (≥99.9% purity and 0.791 g cm$^{-3}$ density) and water (1000 g cm$^{-3}$ density and pH 7) were purchased from Fisher Scientific, Pittsburgh, PA, USA.

### 2.3. Production of Different Biochars

A total of twelve biochars were produced from six feedstocks which included Australian pine (*Casaurina equisetifolia*), Brazilian pepper *(Schinus terebinthifolius)*, coconut husk (*Cocos nusifera*), cypress (*Taxodium distichum*), loblolly pine (*Pinus taeda*) and pecan shell (*Carya illinoinensis*) at temperatures of 350 °C and 500 °C. Australian pine and Brazilian pepper are invasive species in South Florida and have been listed on the Florida's Noxious Weeds List [24]. The biochars were pyrolyzed at the USDA-ARS station in Florence, South Carolina, USA. The biochars were denoted based on the feedstock and production temperature, such as AP350, which indicated Australian pine-derived biochar pyrolyzed at 350 °C or AP500, which indicated Australian pine-derived biochar pyrolyzed at 500 °C, etc.

Biochar yield, defined as the amount of biochar produced at each pyrolysis temperature, was calculated as:

$$Biochar\ Yield\ (\%) = \frac{M_{Biochar}}{M_{Feedstock}} \times 100 \qquad (1)$$

where $M_{Biochar}$ is the mass (g) of biochar and $M_{Feedstock}$ is the mass (g) of feedstock, both on a dry weight basis.

Proximate analysis, a combination of moisture, volatile matter (VM), ash and fixed carbon (C) content of the biochar, used to measure char quality, was conducted following the method of ASTM proximate analysis for wood charcoals [25] in the laboratories at FIU. Moisture and VM content were measured as mass lost at temperatures of 105 °C and 950 °C, respectively, and ash content was measured as the mass remaining after heating at 750 °C using a Fisher Scientific isotemp muffle furnace. Fixed C, which corresponds to the stable carbon fraction of the sample, was determined as:

$$Fixed\ C,\ (\%) = [1 - (ash\ content\ +\ VM\ content)]\ \times\ 100 \tag{2}$$

The pH of the biochar samples was measured in a 1:20 (*w/v*) biochar/deionized water mixture using a Denver instrument pH meter. The cation exchange capacity (CEC) was analyzed by a modified $NH_4^+$ acetate compulsory displacement method [26] using a Perkin Elmer inductively coupled plasma–optical emission spectroscopy (ICP–OES) instrument. Biochar samples were sent to Galbraith Laboratories, Inc., Knoxville, TN, USA for elemental (carbon, hydrogen, nitrogen, oxygen and sulfur content) analyses. The Micromeritics Tristar II surface area and porosity analyzer (Micromeritics, GA, USA) at the Department of Mechanical and Materials Engineering, FIU was used to measure the Brunauer–Emmett–Teller (BET) surface area, pore volume and average pore size of the biochars.

### 2.4. Adsorption and Desorption Experiments

Adsorption of atrazine in soil with and without biochar amendments was measured using the batch equilibrium method [27]. A method by Garcia-Jaramillo et al. [28] was followed for the adsorption and desorption experiments with some modifications. Ten milliliters of atrazine solutions with initial concentrations ($C_i$, mg L$^{-1}$) ranging from 1 to 15 mg L$^{-1}$ were added to 50 mL centrifuge tubes containing 5.0 g of unamended soil and soils amended with 2% (*w/w*) of the twelve biochars. Suspensions were shaken at 120 rpm for 24 h in a platform shaker at 20 ± 2 °C and centrifuged at 1500 rpm for 20 min. The supernatants were filtered through a 0.45 μm filter paper (glass fiber) using a syringe and stored at 4 °C until analyzed. Desorption studies were carried out to evaluate any further removal of atrazine following adsorption by the unamended and biochar-amended soils. Desorption experiments were carried out after adsorption using the samples that had the maximum initial pesticide concentration (15 mg L$^{-1}$) by replacing half of the supernatant solution with deionized water. Equilibrium concentrations ($C_e$, mg L$^{-1}$) of atrazine in the supernatants were analyzed using an Agilent 1260 infinity high-performance liquid chromatography (HPLC) instrument. The calibration curve for atrazine was linear ($R^2 = 0.99$) in a 0.5–20 mg L$^{-1}$ concentration range and the limit of detection (LOD) and limit of quantification (LOQ) were 0.4 mg L$^{-1}$ and 1.2 mg L$^{-1}$, respectively. The HPLC instrument was equipped with a diode array detector and a Hypersil Green ENV C18 analytical column (150 × 4.6 mm, 3 μm). The mobile phase consisted of a methanol/water (50:50, *v/v*) mixture and the detector was set at 222 ± 2 nm. All samples were prepared in triplicate.

The amount of pesticide adsorbed was calculated as

$$C_s\ =\ (C_i - C_e)\ \times\ \frac{V}{M} \tag{3}$$

where *V* = the volume of pesticide solution added (mL).

*M* = mass of adsorbent (g).

The amount of pesticide desorbed was calculated as the difference between the amount of pesticide determined in the solution after the desorption experiment and the amount of pesticide remaining from the adsorption experiment.

The percentage of pesticide adsorbed was calculated as

$$\% \ Adsorption \ = \ \frac{C_i - C_e}{C_i} \ \times \ 100 \tag{4}$$

The percentage of pesticide desorbed was calculated as the ratio between the amount of pesticide desorbed and the amount adsorbed at equilibrium.

All the adsorption and desorption isotherms were fitted using the Freundlich equation:

$$\log C_s = \log K_f + \frac{1}{n} \log C_e \tag{5}$$

where $C_s$ = amount of pesticide adsorbed (mg kg$^{-1}$).

$C_e$ = equilibrium concentration of pesticide (mg L$^{-1}$).

$K_f$ and $1/n$ are empirical constants.

$K_f$ is the sorption coefficient which indicates the sorption capacity of pesticide and $1/n$ is the slope isotherm which reflects the sorption intensity ($1/n = 1$ represents a linear isotherm curve). The values of $K_f$ cannot be compared due to variation in $1/n$ values. Therefore, $K_d$ was the calculated ratio between the amount of sorbed pesticide and the equilibrium concentration of 1 mg L$^{-1}$. The estimated $K_d$ values were further normalized to the organic carbon (OC) content of the soil to quantify $K_{OC}$ values:

$$K_{OC} = \frac{K_d}{\%OC} \ \times \ 100 \tag{6}$$

The hysteresis coefficient was determined as

$$H = \ \frac{1}{n_{des}} \bigg/ \frac{1}{n_{ads}} \tag{7}$$

which gives information about the reversibility of adsorption.

### 2.5. Statistical Analyses

All data were presented as means with standard errors. Statistical analysis was performed using a statistical analysis system (SAS 9.4 and JMP Pro v.14). The mean values were examined by one-way analysis of variance (ANOVA) using the SAS 9.4 PROCMIXED procedure. Tukey–Kramer post hoc tests were performed to compare mean separation at $p < 0.05$ among treatments. Any differences between the mean values at $p < 0.05$ were considered statistically significant. Principal component analysis (PCA) was conducted on different biochar characterization parameters using JMP Pro v.14 and hierarchical cluster analysis was done by the centroid method. Regression analysis was performed on adsorption and desorption isotherms.

## 3. Results and Discussion

### 3.1. Physiochemical Properties of Soil and Biochars

The soil used in this study had high OM content (>15%) possibly because the soil was collected from an organic garden which had compost incorporated and residues from a previous cropping season (Table 1). The soil was slightly alkaline (7.52) with a loamy sand texture. The high soil pH was not expected considering the soil parent material is of marine origin and consisted of calcareous material.

Detailed information on biochar yield, proximate analysis and physiochemical properties are presented in Tables 2 and 3 which show considerable variation between the twelve different biochars used in this study. Yield (%), VM (%) and moisture (%) content of the biochars produced from pyrolysis of the six different feedstocks (Australian pine, AP; Brazilian pepper, BP; coconut husk, CH; cypress, Cy; loblolly pine, L; and pecan shell, P) decreased significantly ($p < 0.05$) as the temperature increased from 350 °C to 500 °C (Table 2). The average yield reduction (about 18%) of the biochars with increasing

pyrolysis temperature was due to the dehydration of hydroxyl ($OH^-$) groups and thermal degradation of ligno-cellulose structures [29]. Average VM and moisture contents were reduced by 18% and 50%, respectively, with increasing pyrolysis temperature. Higher pyrolysis temperature can increase the degree of aromatization [30] and cause a greater loss of gas products, tar oil and low molecular weight hydrocarbons (such as methane ($CH_4$), ethane ($C_2H_6$), and propane ($C_3H_8$)) [31] which potentially reduce the VM contents of the biochars produced. The reduction in VM was an indication of greater pore formation on the biochars at higher pyrolysis temperatures [32]. Pores produced in high-temperature biochars serve as a potential habitat for microorganisms [33,34] which eventually increase the sorption ability of organic compounds in the soil [35].

The ash content of the biochars significantly ($p < 0.05$) increased with increasing pyrolysis temperature, probably because the ash mainly remains in the solid fraction and increasing temperature increases the concentrations of minerals and combusted organic residues [36]. An increase of more than double in the ash content of the biochars was observed as the pyrolysis temperature increased from 350 °C to 500 °C (Table 2). Ash is an important factor that influences the sorption behavior of hydrophobic organic compounds (HOCs) which can block surface sorption sites in biochar or make it difficult to access due to their interactions with inorganic moieties [21,37].

Similar to ash content, an increase in pyrolysis temperature also significantly increased the fixed C content of the biochars. An increase in pyrolysis temperature from 350 °C to 500 °C had resulted in about a 70% increase (significant at $p < 0.05$) in fixed C content of the biochars (Table 2) mainly because a higher pyrolysis temperature can reduce overall biochar mass [38].

An average of a 10% increase of biochar pH with increased pyrolysis temperature possibly resulted from the gradual removal of acid functional groups (such as carboxylic (-COOH), phenolic ($-C_6H_5$) and carbonyl (-C=O) groups) from the biochar surface and a relative increase in ash contents. Biochars with pH in the alkaline range have the potential for neutralizing or increasing the pH of acidic soils [21] which, in turn, provides a more favorable habitat for plants and microbes [39–41]. A study conducted by Novak et al. [29] at South Carolina, USA showed that an application of 2% pecan shell derived biochar (produced at 700 °C) significantly increased the soil pH from 4.8 to 6.3.

Cation exchange capacity (CEC) of the biochars decreased with increased pyrolysis temperature, however, no significant difference was observed (Table 2). As discussed earlier, an increase in temperature resulted in the loss of oxygen-containing groups, such as hydroxyl ($OH^-$), carboxylic (-COOH), and carbonyl (-C=O) groups, which resulted in the decrease in the biochar CEC [31,42]. The CEC was also found to be associated with O/C ratios (Table 3) where a higher O/C ratio produced a higher CEC value. In a recent study conducted in Brazil, Batista et al. [42] used *Cocos nusifera* (coconut shell), *Citrus sinensis* (orange peel), *Elaeis guineensis* (palm oil bunch), *Saccharum officinarum* (sugarcane bagasse) and *Eichhornia crassipes* (water hyacinth) feedstocks to make biochars at 350 °C and found that lower O/C ratios are associated with lower CEC of the biochars. The CEC of biochars has the potential to retain nutrients in the soil [31]. High-CEC biochars can also be beneficial for the remediation of cationic trace elements found in contaminated soil [43,44].

**Table 2.** Biochar yield, proximate analysis, and selected physicochemical properties of the twelve different biochars.

| Sample [†] | Feedstock | Production Temperature (°C) | Biochar Yield (%) | Volatile Matter Content [‡] (%) | Ash Content [§] (%) | Moisture Content [¶] (%) | Fixed C [#] (%) | pH [††] | CEC [‡‡] (cmol kg$^{-1}$) | SSA [§§] (m$^2$ g$^{-1}$) | TPV [¶¶] (cm$^3$ g$^{-1}$) | Average Pore Size (nm) |
|---|---|---|---|---|---|---|---|---|---|---|---|---|
| | | | | Proximate Analysis | | | | | | | | |
| AP350 | Australian pine | 350 | 41.00 | 79.24 ± 2.39 AB | 5.32 ± 1.11 B | 5.85 ± 0.09 ABCD | 15.44 ± 3.50 B | 8.58 ± 0.03 C | 16.31 ± 5.50 A | 0.98 ± 0.07 | 0.003 | 12.46 |
| AP500 | (*Casaurina equisetifolia*) | 500 | 33.10 | 61.35 ± 5.67 ABC | 10.19 ± 0.91 A | 2.81 ± 0.19 DE | 28.46 ± 6.59 AB | 9.37 ± 0.03 B | 8.19 ± 1.43 A | 2.59 ± 0.29 | 0.006 | 9.40 |
| BP350 | Brazilian pepper | 350 | 41.60 | 66.47 ± 9.32 ABC | 2.06 ± 1.00 CD | 4.40 ± 0.47 BCDE | 31.48 ± 8.32 AB | 7.72 ± 0.08 DE | 8.47 ± 2.51 A | 0.57 ± 0.08 | 0.002 | 12.26 |
| BP500 | (*Schinus terebinthifolius*) | 500 | 33.00 | 55.80 ± 3.79 BC | 4.02 ± 0.02 BC | 1.96 ± 0.98 E | 40.18 ± 3.82 AB | 9.65 ± 0.02 AB | 7.92 ± 2.30 A | 2.29 ± 0.26 | 0.008 | 14.60 |
| CH350 | Coconut husk | 350 | 47.20 | 85.05 ± 2.45 A | 3.74 ± 0.48 BC | 8.81 ± 0.81 A | 11.21 ± 1.97 B | 9.40 ± 0.09 B | 16.32 ± 3.46 A | 0.89 ± 0.15 | 0.003 | 13.31 |
| CH500 | (*Cocos nusifera*) | 500 | 40.30 | 79.37 ± 1.48 AB | 8.88 ± 0.38 A | 4.96 ± 0.92 BCDE | 11.75 ± 1.85 B | 9.89 ± 0.10 A | 12.04 ± 1.07 A | 1.94 ± 0.22 | 0.004 | 7.99 |
| Cy350 | Cypress | 350 | 37.70 | 72.75 ± 1.17 ABC | 0.55 ± 0.05 D | 6.51 ± 0.57 ABC | 26.71 ± 1.22 AB | 7.11 ± 0.01 G | 10.55 ± 0.20 A | 0.41 ± 0.07 | 0.001 | 10.01 |
| Cy500 | (*Taxodium distichum*) | 500 | 30.00 | 62.66 ± 7.56 ABC | 1.59 ± 0.11 CD | 2.45 ± 0.52 E | 36.76 ± 7.67 AB | 7.67 ± 0.01 DE | 9.18 ± 2.46 A | 4.18 ± 0.47 | 0.002 | 2.39 |
| L350 | Loblolly pine | 350 | 39.60 | 71.31 ± 6.00 ABC | 1.70 ± 0.10 CD | 3.46 ± 0.46 CDE | 26.98 ± 6.11 AB | 7.63 ± 0.05 EF | 8.51 ± 1.84 A | 0.30 ± 0.06 | 0.001 | 12.81 |
| L500 | (*Pinus taeda*) | 500 | 32.20 | 48.59 ± 7.53 C | 3.20 ± 0.19 BCD | 2.36 ± 0.56 E | 48.21 ± 7.73 A | 7.84 ± 0.01 DE | 7.93 ± 4.34 A | 5.21 ± 0.56 | 0.004 | 3.13 |
| P350 | Pecan shell | 350 | 46.80 | 68.04 ± 4.12 ABC | 2.18 ± 0.12 CD | 6.83 ± 0.09 AB | 29.78 ± 4.24 AB | 7.36 ± 0.02 FG | 6.14 ± 1.18 A | 0.36 ± 0.05 | 0.001 | 14.56 |
| P500 | (*Carya illinoinensis*) | 500 | 39.20 | 56.33 ± 1.12 ABC | 3.82 ± 0.29 BC | 3.50 ± 0.50 CDE | 39.85 ± 1.42 AB | 7.94 ± 0.03 D | 4.66 ± 1.41 A | 2.14 ± 0.34 | 0.002 | 4.41 |

[†] Sample abbreviations are as follows: AP350 & AP500 = Australian pine derived biochar pyrolyzed at 350 °C and 500 °C, BP350 & BP500 = Brazilian pepper derived biochar pyrolyzed at 350 °C and 500 °C, CH350 & CH500 = Coconut husk derived biochar pyrolyzed at 350 °C and 500 °C, Cy350 & Cy500 = Cypress derived biochar pyrolyzed at 350 °C and 500 °C, L350 & L500 = Loblolly pine derived biochar pyrolyzed at 350 °C and 500 °C, P350 & P500 = Pecan shell derived biochar pyrolyzed at 350 °C and 500 °C; [‡‡] CEC = Cation exchange capacity; [§§] SSA = Specific surface area; [¶¶] TPV = Total pore volume. Numbers are mean ± standard error; similar letter in the table indicates not significantly different at $p < 0.05$.

**Table 3.** Elemental composition and atomic ratio of the twelve different biochars.

| Sample [†] | Carbon (%) | Hydrogen (%) | Nitrogen (%) | Oxygen (%) | Sulfur (%) | Atomic Ratio of the Elements in Biochar | | |
| | | | | | | H/C [‡] | O/C [§] | (N+O)/C [¶] |
|---|---|---|---|---|---|---|---|---|
| AP350 | 64.93 | 4.00 | 0.94 | 21.80 | 0.07 | 0.06 | 0.34 | 0.35 |
| AP500 | 66.65 | 3.07 | 1.10 | 15.66 | 0.04 | 0.05 | 0.23 | 0.25 |
| BP350 | 67.54 | 3.97 | 0.5 | 20.89 | 0.11 | 0.06 | 0.31 | 0.32 |
| BP500 | 77.37 | 3.04 | 0.51 | 11.76 | 0.13 | 0.04 | 0.15 | 0.16 |
| CH350 | 66.69 | 4.02 | 0.51 | 22.81 | 0.03 | 0.06 | 0.34 | 0.35 |
| CH500 | 67.00 | 3.01 | 0.69 | 18.29 | 0.03 | 0.04 | 0.27 | 0.28 |
| Cy350 | 76.10 | 4.39 | 0.5 | 17.31 | 0.01 | 0.06 | 0.23 | 0.23 |
| Cy500 | 83.59 | 3.39 | 0.5 | 11.11 | 0.04 | 0.04 | 0.13 | 0.14 |
| L350 | 67.71 | 4.25 | 0.5 | 17.13 | 0.09 | 0.06 | 0.25 | 0.26 |
| L500 | 79.47 | 3.52 | 0.5 | 12.97 | 0.08 | 0.04 | 0.16 | 0.17 |
| P350 | 68.45 | 3.59 | 0.5 | 22.85 | 0.04 | 0.05 | 0.33 | 0.34 |
| P500 | 78.96 | 3.21 | 0.5 | 12.10 | 0.03 | 0.04 | 0.15 | 0.16 |

[†] Sample abbreviations are the same as in Table 2; [‡] H/C = ratio of hydrogen and carbon; [§] O/C = ratio of oxygen and carbon; [¶] (N+O)/C = ratio of nitrogen and oxygen with carbon.

Specific surface area (SSA) and total pore volume (TPV) of the biochars were numerically increased (but not significant at $p < 0.05$) with higher temperatures. However, the average pore size of different biochars decreased with increased charring temperature with the exception of biochars made from BP where the average pore size increased at 500 °C. The decomposition of cellulose and hemicelluloses, removal of pore-blocking substances, destruction of aliphatic alkyls and ester groups, exposure of the aromatic lignin core, thermal cracking and formation of vesicular bundles or channel structures have been attributed to the higher SSA and TPV in biochars produced in higher pyrolysis temperatures [45,46]. The condensation reaction of organic compounds causes a decrease in biochar pore size with increasing pyrolysis temperature [47]. Biochars with higher SSA and TPV are considered as potential agents for the sorption of organic compounds in soil [35].

The biochars had both carbonized (such as volatile matter, carbon content, H/C, O/C ratio) and non-carbonized (such as ash content) fractions (Tables 2 and 3), indicating that the biochars were heterogeneous. It was observed that the C (%) content of the biochars increased while the H (%) and O (%) contents decreased as the pyrolysis temperature increased from 350 °C to 500 °C (Table 3). The increase in C content was due to the high carbonization and high degree of carbon clustering in the aromatic structures as a result of an increase in temperature [29,48]. The reduction in H content was relatively small due to an increase in temperature and almost negligible for the different feedstocks. All the biochars contained a relatively small amount of N (%) which ranged from 0.5% to 1.1% and the N content remained relatively stable with very little change, regardless of temperature and feedstock. The elemental composition of the biochars was used to calculate the atomic ratio for each biochar (Table 3). The H/C, O/C and (N+O)/C ratios for all biochars decreased as the pyrolysis temperature increased. The elemental ratio of H/C is used to evaluate the degree of carbonization and aromaticity of the biochar and is linked to the long-term stability in the environment [49]. The lower values represent a high degree of carbonization and aromaticity. A decrease in O/C and (N+O)/C ratios indicates a reduction in biochar polarity. An increase in aromaticity and a decrease in polarity reflect a higher sorption capacity of the biochar.

Coconut husk (CH) biochars had a higher yield (1–10%), VM content (12–21%), moisture content (3.5–9.5%), pH (8–30%) and CEC (16–163%) than the biochars made from the remaining five feedstocks. Comparing the 12 biochars, it was found that CH350 biochar had the highest (significant at $p < 0.05$) VM (85%) and moisture content (8.81%) (Table 2). A higher biochar yield from coconut husk (CH), which can be considered as an agricultural residue (or waste), makes it an excellent feedstock for manufacturing abundant biochar in a cost-effective way. The high pH (9.40 to 9.89) of CH biochars makes them useful as a liming

material in acidic soils and the higher CEC (12.02 to 16.32 cmol kg$^{-1}$) can be effective in removing inorganic and organic contaminants from the soil.

The principal component analysis (PCA) biplot of biochar properties is presented in Figure 1. Principal component 1 (PC1) and principal component 2 (PC2) explain 48% and 21% of the total variance of the results. For PC1, the main contributing parameters were O content, O/C and (N+O)/C ratios, whereas pH, ash content, TPV, H content and N content were the main contributing parameters for PC2. The O/C and (N+O)/C ratios are associated with the polarity of the biochars, influencing their sorption capacity, which is reflected in the adsorption–desorption experiment.

The parameters from the PCA were also used for hierarchical cluster analysis of the biochars (Figure 2). Biochars made at 500 °C were different from the ones made at 350 °C. Among biochars made at 500 °C, AP500 and CH500 were clustered together, reflecting maximum similarity. Biochars made at 350 °C, Cy350 and L350 showed more similarity than the remaining biochars. This type of clustering is helpful in further explaining the similar effects of the biochars during the adsorption and desorption of atrazine.

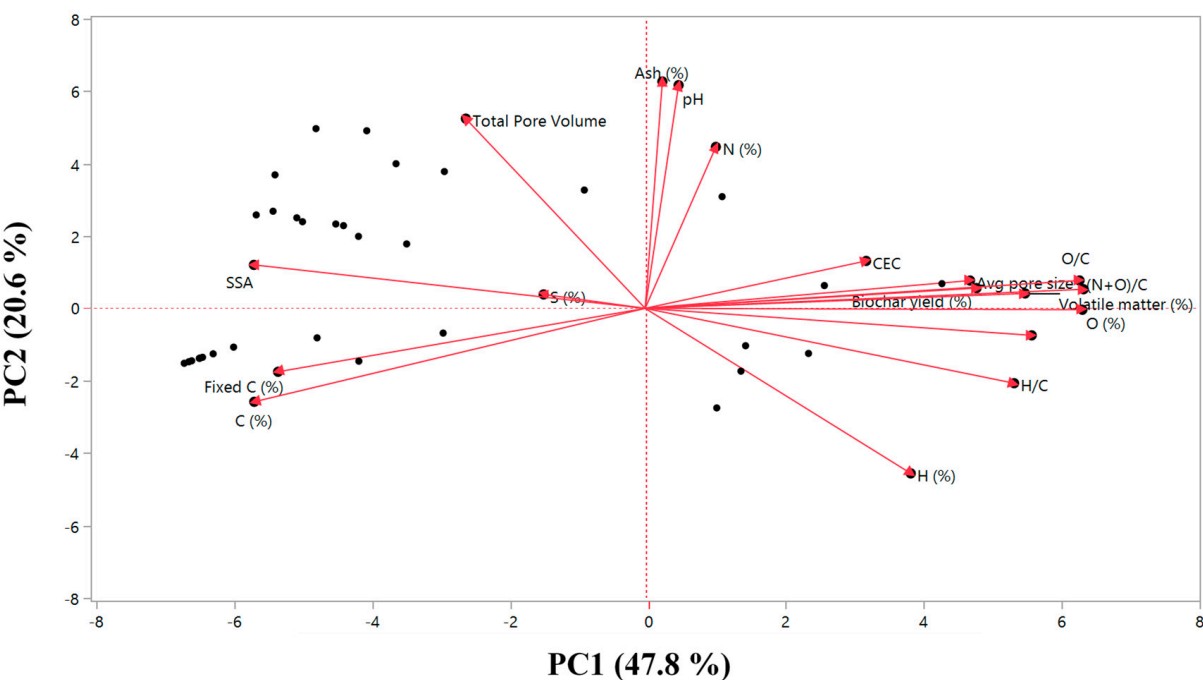

**Figure 1.** Principal component analysis (PCA) biplot of the parameters measured for the characterization of biochars. PC1 explains 48% of the variance and is mainly a combination of O content (%), O/C and (N+O)/C ratios. PC2 is mainly a combination of pH and ash content (%) and explains 21% of the variance.

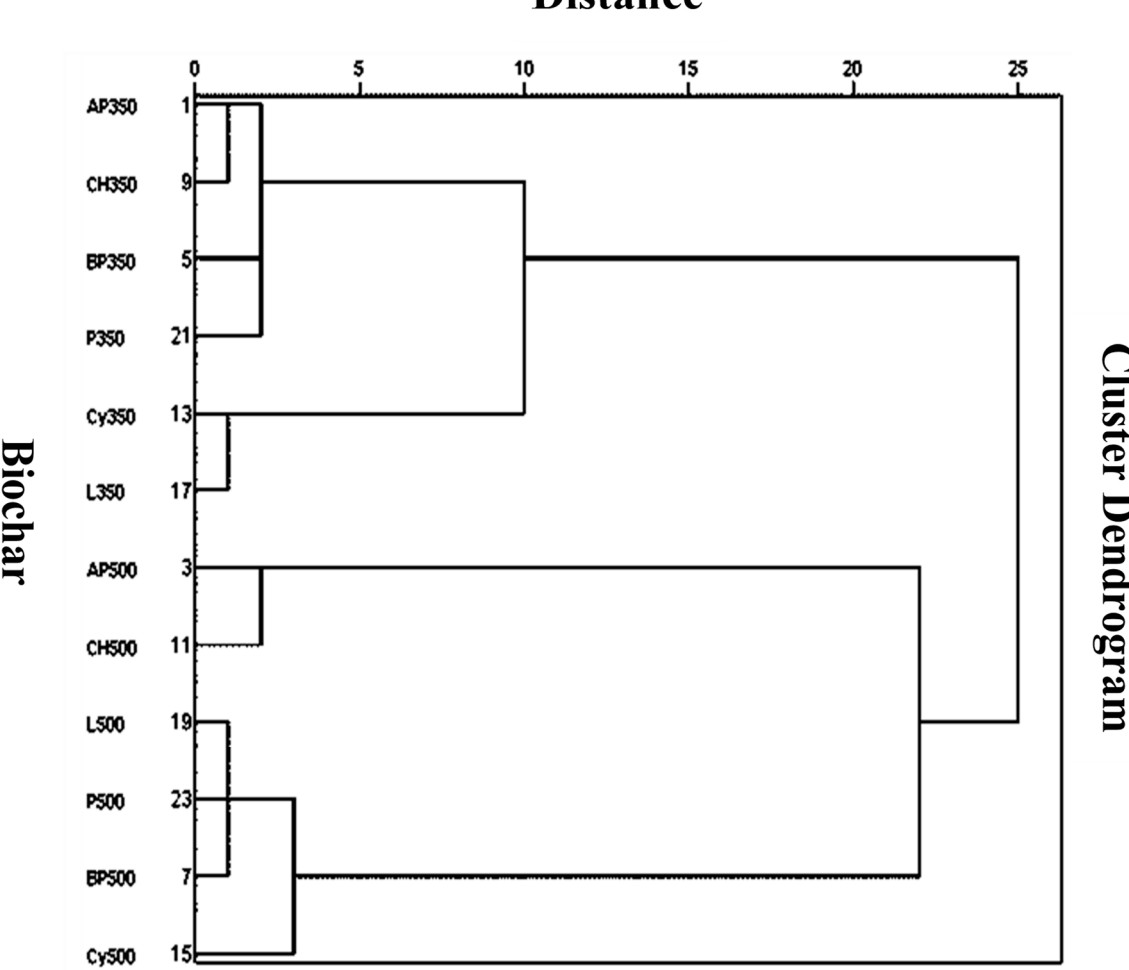

**Figure 2.** Hierarchical cluster analysis of the biochars based on the parameters from principal component analysis. Biochars with similar characteristics are clustered together. Biochar abbreviations are the same as in Table 2.

### 3.2. Adsorption–Desorption Isotherms of Atrazine

The Freundlich adsorption–desorption isotherm was used in this study to describe partitioning of atrazine between the biochar/soil solution and in the solid surface. The adsorption–desorption isotherms (Figures 3 and 4) helped us understand the nature of interactions between atrazine and soils with and without biochar amendments. The adsorption isotherms fit to the Freundlich equation well. The correlation coefficients, $r^2$, ranged from 0.96 to 0.99 for soils amended with AP350, CH350, CH500, Cy350, Cy500, L350, L500, and P350 biochars and $r^2$ was from 0.73 to 0.77 for soils amended with AP500, BP350, BP500, and P500 biochars.

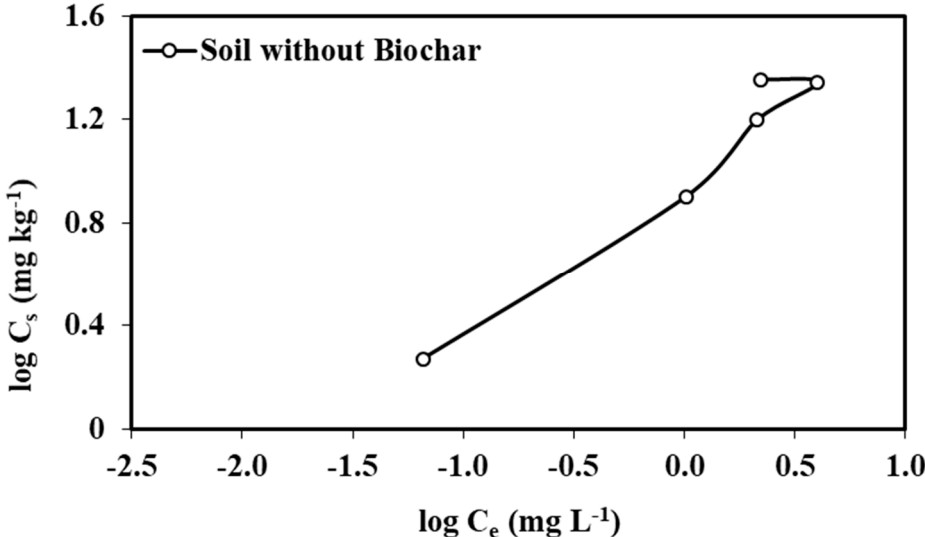

**Figure 3.** The adsorption–desorption isotherms for atrazine calculated in soil without biochar.

The $1/n_{ads}$ values of atrazine adsorption in the unamended soil and biochar-amended soils were less than 1, suggesting that the adsorption isotherms were nonlinear and L-type (Table 4; Figures 3 and 4). The L-type isotherms generally indicate that adsorption is strongly dependent on the initial solution concentration and there is a decrease in adsorption at higher solution concentrations of the pesticide [28,35]. An increase in the degree of isotherm nonlinearity and L-type isotherms reflected that pore filling was the primary mechanism for the sorption of atrazine by the biochar-amended soils [28,35]. Soils amended with biochars made from cypress (Cy), loblolly pine (L) and pecan shell (P) at 350 °C had $1/n_{ads}$ values greater (0.60, 0.60 and 0.60, respectively) than that of unamended soil (0.59). However, all other biochar-amended soils had $1/n_{ads}$ values lower than that of the unamended soil, suggesting that these biochars had higher affinity for atrazine which resulted from a condensed sorption domain [21].

**Table 4.** Adsorption parameters for atrazine in unamended soil and soil amended with the twelve different biochars at 2% (*w*/*w*).

| Treatment [†] | $K_{f\ ads}$ [‡] | $1/n_{ads}$ [§] | Freundlich $r^2$ | $K_{d\ ads}$ [¶] | $K_{OC}$ [#] | %Adsorption |
|---|---|---|---|---|---|---|
| Soil | 9.12 ± 1.07 | 0.59 ± 0.04 | 0.99 | 9.12 [BCDE] | 101.53 [BCDE] | 73.24–93.44 (81.25) [††] |
| Soil + AP350 | 10.47 ± 1.15 | 0.47 ± 0.06 | 0.96 | 10.47 [ABCDE] | 116.56 [ABCDE] | 75.77–97.63 (83.09) |
| Soil + AP500 | 9.17 ± 1.48 | 0.43 ± 0.14 | 0.77 | 9.17 [E] | 102.09 [E] | 58.72–98.96 (79.20) |
| Soil + BP350 | 8.32 ± 1.45 | 0.42 ± 0.19 | 0.72 | 8.32 [CDE] | 92.63 [CDE] | 52.26–98.28 (76.73) |
| Soil + BP500 | 9.33 ± 1.45 | 0.36 ± 0.15 | 0.73 | 9.33 [CDE] | 103.87 [CDE] | 54.51–98.73 (77.72) |
| Soil + CH350 | 13.80 ± 1.02 | 0.52 ± 0.02 | 0.99 | 13.80 [A] | 153.63 [A] | 80.96–100.0 (90.31) |
| Soil + CH500 | 10.96 ± 1.02 | 0.54 ± 0.02 | 0.99 | 10.96 [ABC] | 122.02 [ABC] | 75.03–100.0 (87.05) |
| Soil + Cy350 | 7.94 ± 1.05 | 0.60 ± 0.03 | 0.99 | 7.94 [DE] | 88.40 [DE] | 71.35–93.99 (78.52) |
| Soil + Cy500 | 11.40 ± 1.07 | 0.40 ± 0.03 | 0.99 | 11.40 [ABCDE] | 126.92 [ABCDE] | 71.32–98.90 (83.97) |
| Soil + L350 | 11.39 ± 1.12 | 0.60 ± 0.07 | 0.98 | 11.39 [BCDE] | 126.80 [BCDE] | 80.10–95.48 (85.42) |
| Soil + L500 | 12.30 ± 1.15 | 0.33 ± 0.05 | 0.96 | 12.30 [ABCD] | 136.93 [ABCD] | 76.84–99.65 (85.12) |
| Soil + P350 | 9.12 ± 1.02 | 0.60 ± 0.02 | 0.99 | 9.12 [CDE] | 101.53 [CDE] | 72.24–93.01 (80.82) |
| Soil + P500 | 13.49 ± 1.48 | 0.52 ± 0.21 | 0.75 | 13.49 [AB] | 150.18 [AB] | 78.45–97.18 (87.53) |

Treatment [†] = Treatment abbreviations are the same as in Table 2; [‡] $K_{f\ ads}$ = Freundlich sorption coefficient; [§] $1/n_{ads}$ = Freundlich slope constant; [¶] $K_{d\ ads}$ = sorption coefficient estimated from the Freundlich sorption isotherms at equilibrium concentration ($C_e$) of 1.0 mg L$^{-1}$; [#] $K_{OC} = (K_{d\ ads}/\%OC) \times 100$, sorption coefficient ($K_{d\ ads}$) normalized to the organic carbon (OC) content of the soil; [††] = number in parentheses is the average adsorption by each treatment across the entire range of pesticide concentrations; numbers are mean ± standard error; means within a column followed by the same letter are not significantly different at $p < 0.05$.

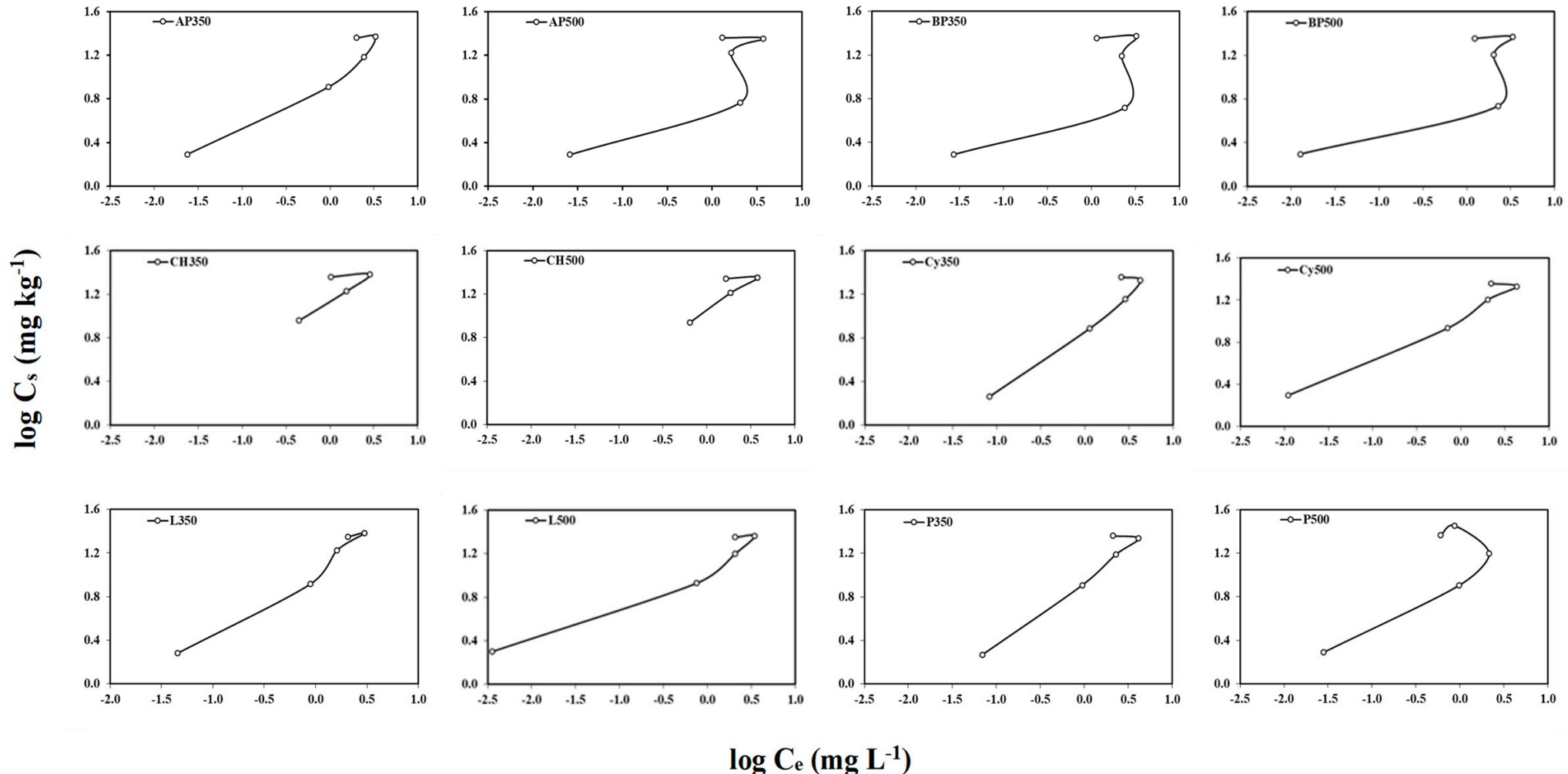

**Figure 4.** The adsorption–desorption isotherms for atrazine calculated in soil amended with the twelve different biochars at 2% (*w*/*w*).

Interestingly, pyrolysis temperature did not have a significant effect on atrazine adsorption among different feedstocks. The results suggest that coconut husk (CH) had the highest adsorption (%) of atrazine among the six feedstocks, followed by loblolly pine (L) and pecan shell (P) (Table 4). Only Brazilian pepper (BP) feedstock had lower (5% but not significant at $p < 0.05$) atrazine adsorption than unamended soil for this experiment, which was most likely due to the very low CEC that was observed in biochars from BP feedstocks (Table 2). Overall, the atrazine adsorption capacity of biochars made from loblolly pine and pecan shell (native to Southeastern US) was 7–10% higher than biochars made from Australian pine and Brazilian pepper (non-native to Florida and invasive species), indicating the possibility that biochars produced from native plant species have comparative advantages of retaining atrazine in the same agroclimatic regions where they grow. Biochars from loblolly pine and pecan shell have been documented as effective adsorbents of organic compounds in previous studies [50,51]. Among all 12 biochars, CH350 performed best and had about 1.2 times higher ($p < 0.05$) adsorption than unamended soil, possibly resulting from the highest CEC of CH350 (16.32 cmol kg$^{-1}$). Coconut trees are very common in the tropical climate of Florida, USA and, specifically in South Florida (due to the shoreline and sea beaches), the abundance is much higher than other parts of the state. Therefore, it was assumed that biochar production from coconut husk in South Florida would be an effective solution to atrazine adsorption for other agricultural settings in the US.

The biochar amendments significantly influenced the adsorption of atrazine compared to the unamended soil. Overall, the $K_{d\,ads}$ values, which indicate the sorption affinity for atrazine (high value means higher affinity) were higher in biochars made from AP, CH and L at 350 °C and 500 °C, and from BP, Cy and P at 500 °C than the unamended soil (Table 4), most likely because of the higher SSA, TPV and aromaticity (low H/C ratio) of those biochars. Lower $K_{d\,ads}$ values of biochars made from BP and Cy at 350 °C than the unamended soil suggests that these biochars may not be a viable option for atrazine adsorption in agricultural soils. Since soil OC is considered to be the primary soil component controlling the sorption of pesticide, the $K_{d\,ads}$ values were further normalized to $K_{OC}$ values for predicting better sorption behaviors of atrazine Equation (6). This normalization assumes that OM is the primary soil property controlling adsorption, otherwise, there will be variation in the Koc values [52]. Greater $K_{OC}$ values of biochar-amended soils than that of the unamended soil suggest that the biochars exhibit higher sorptivity to pesticide than the soil OC [53]. The $K_{OC}$ values of soils amended with biochars made from AP, CH and L at 350 °C and 500 °C, and from BP, Cy and P at 500 °C were greater than that of the unamended soil (Table 4), emphasizing a greater affinity for atrazine of these biochars than the soil OM. Out of the 12 biochars, the $K_{OC}$ value for CH350 was significantly ($p < 0.05$) higher than the unamended soil. The $K_{OC}$ values of atrazine adsorption to biochars were in the range of 1 to 1.5 times higher than adsorption to soil. The addition of biochars in soils may have resulted in the increase in OC in soil, which, in turn, enhanced the adsorption of atrazine in the biochar-amended soils. Application of biochars made from CH and L at 350 °C and 500 °C, from AP at 350 °C and from Cy and P at 500 °C increased the overall percentage of adsorption of atrazine in soil, with CH350 biochar-amended soil having the highest value (Table 4).

Desorption isotherms were also adequately described by the Freundlich adsorption–desorption equation, indicated by the correlation coefficient ($r^2$) values that ranged between 0.62 to 0.97 (Table 5). Larger $K_{d\,des}$ values indicate that a greater proportion of the pesticide is retained by the biochar-amended soils following the desorption experiment [52,54]. Overall $K_{d\,des}$ values for atrazine were 16% higher than $K_{d\,ads}$ values, showing that a considerable amount of the atrazine that was adsorbed by the biochar-amended soils were retained following the desorption experiment. Since the $K_{d\,des}$ values of soils amended with biochars made from AP, CH and L at 350 °C and 500 °C, and from BP and P at 500 °C, were higher than the unamended soil (Table 5), it can be assumed that these biochars retained a larger amount of adsorbed atrazine than the unamended soil.

**Table 5.** Desorption parameters for atrazine in unamended soil and soil amended with the twelve different biochars at 2% (*w/w*).

| Treatment [†] | $K_{f\,des}$ [‡] | $1/n_{des}$ [§] | Freundlich $r^2$ | $K_{d\,des}$ [¶] | H [#] | %Desorption |
|---|---|---|---|---|---|---|
| Soil | $10.00 \pm 1.22$ | $0.62 \pm 0.07$ | 0.96 | 10.00 | 1.07 [A] | 3.52 |
| Soil + AP350 | $11.48 \pm 1.15$ | $0.49 \pm 0.07$ | 0.94 | 11.48 | 1.04 [AB] | 2.69 |
| Soil + AP500 | $10.72 \pm 1.35$ | $0.45 \pm 0.17$ | 0.70 | 10.72 | 1.05 [B] | 2.30 |
| Soil + BP350 | $10.00 \pm 1.41$ | $0.43 \pm 0.19$ | 0.62 | 10.00 | 1.02 [AB] | 3.19 |
| Soil + BP500 | $10.96 \pm 1.38$ | $0.38 \pm 0.16$ | 0.66 | 10.96 | 1.06 [A] | 3.12 |
| Soil + CH350 | $16.98 \pm 1.12$ | $0.48 \pm 0.18$ | 0.70 | 16.98 | 0.92 [AB] | 2.11 |
| Soil + CH500 | $15.85 \pm 1.09$ | $0.54 \pm 0.11$ | 0.79 | 15.85 | 1.00 [AB] | 6.03 |
| Soil + Cy350 | $8.71 \pm 1.12$ | $0.64 \pm 0.08$ | 0.95 | 8.71 | 1.07 [AB] | 2.82 |
| Soil + Cy500 | $12.30 \pm 1.09$ | $0.42 \pm 0.04$ | 0.97 | 12.30 | 1.05 [AB] | 2.66 |
| Soil + L350 | $12.02 \pm 1.09$ | $0.61 \pm 0.06$ | 0.97 | 12.02 | 1.02 [AB] | 4.94 |
| Soil + L500 | $13.49 \pm 1.12$ | $0.35 \pm 0.05$ | 0.95 | 13.49 | 1.06 [AB] | 4.45 |
| Soil + P350 | $9.77 \pm 1.12$ | $0.63 \pm 0.08$ | 0.96 | 9.77 | 1.05 [AB] | 2.26 |
| Soil + P500 | $15.85 \pm 1.41$ | $0.53 \pm 0.21$ | 0.68 | 15.85 | 1.02 [AB] | 1.20 |

Treatment [†] = Treatment abbreviations are the same as in Table 2; [‡] $K_{f\,des}$ = Freundlich sorption coefficient; [§] $1/n_{des}$ = Freundlich slope constant; [¶] $K_{d\,des}$ = sorption coefficient estimated from the Freundlich sorption isotherms at equilibrium concentration ($C_e$) = 1.0 mg L$^{-1}$; [#] H = $(1/n_{des})/(1/n_{ads})$, hysteresis coefficient; numbers are mean $\pm$ standard error; means within a column followed by the same letter are not significantly different at $p < 0.1$.

The difference between desorption and adsorption isotherms for soils with and without biochars is indicative of hysteresis. The hysteresis index (H), calculated as the ratio between $1/n_{des}$ and $1/n_{ads}$ (Equation (7)), is positive when the value is less than 1 and is negative when it is more than 1. Positive hysteresis was observed in soil amended with coconut husk biochar (CH), where the H value for CH350 biochar-amended soil was less than 1 (Table 5), implying that it was difficult to desorb atrazine that had already been sorbed by this biochar. The hysteresis that occurred may be due to the irreversible binding of atrazine on biochar in sorption sites or due to entrapment in the porous structure of biochar, which made it difficult for the atrazine molecules to be washed out [55,56]. In this study, the $1/n_{des}$ values were slightly higher than the $1/n_{ads}$ values for the unamended soil and soils amended with biochars made from AP, BP, Cy, L and P, resulting in H values higher than 1. Therefore, negative hysteresis was observed in these cases, indicating that the rate of desorption was slightly higher than the rate of adsorption. The percentage of desorption for soil amended with biochars made from CH at 500 °C and from L at 350 °C and 500 °C were higher than the unamended soil (Table 5). The sorption of atrazine is pH dependent, and desorption increases with an increase in pH [21,54]. The pH of CH500 (9.89), L350 (7.63) and L500 (7.84) biochars was higher than the unamended soil (7.52), which may have contributed to the higher percentage of desorption shown by these biochars. The lowest percentage of desorption was observed in P500 biochar-amended soil (1.20%) which had the second highest adsorption (Tables 4 and 5), indicating that this biochar was able to retain the maximum amount of adsorbed atrazine, which can help in reducing further surface and groundwater contamination. Therefore, pecan shell (P), which is considered as an agricultural waste, can be considered an effective feedstock for the mass production of biochar that can be used to reduce atrazine contamination in the environment.

### 3.3. Environmental Implications

Despite causing surface and groundwater contamination, atrazine is still one of the most widely used pesticides in the world. In fact, it is the second most commonly used pesticide in US agriculture and its sale at present is still steadily maintained at 31.75–36.29 million kg per year [57]. The maximum contaminant level (MCL) for atrazine in drinking water established by USEPA is 3.0 μg/L and the European Union requires an MCL below 0.1 μg/L for a single pesticide in drinking water. As atrazine has a half-life of one to twelve months in the environment and can persist in soil for up to a decade,

it has been frequently detected in surface and groundwater resources at concentrations many times above the 0.1 µg L$^{-1}$ groundwater quality standard for individual pesticides or 0.5 µg L$^{-1}$ for the sum of several pesticides [23,58,59]. It is well known that atrazine is a highly mobile toxin to aquatic organisms, plants and human beings [4]. The use of biochar as an adsorbent offers great potential for removing pesticides from the environment in an efficient and cost-effective manner. We found that, despite the high indigenous soil organic matter (SOM), the biochar amendments enhanced the adsorption of atrazine, as indicated by the K$_{OC}$ values. The higher observed adsorption by biochars made from AP, CH and L at 350 and 500 °C, and BP, Cy and P at 500 °C than unamended soil indicates the effectiveness of these biochars in retaining atrazine, thus reducing further surface and groundwater contamination. In general, the high SSA, carbonaceous nature, hydrophobicity and porous structure of the biochar play an important role in effectively influencing the sorption of pesticides. In this study, the CEC of the biochars, in addition to SSA, TPV and aromaticity, influenced the higher adsorption of atrazine. It is evident that biochars produced from different feedstocks have various physiochemical properties that greatly influence the sorption capacity for pesticides. Results from Delwiche et al. [20] using biochar made from pine wood chips pyrolyzed between 300 °C to 550 °C resulted in a decrease in the leaching of atrazine from a homogenized soil column by 52%. Huang et al. [22] conducted a study in China using biochars made from sugarcane at 500 °C, where the addition of biochars increased the adsorption of atrazine by 27% in a moist soil with a low level of total organic carbon, while it increased adsorption by 32% in a paddy soil with high total organic carbon. In both studies, the high adsorption of atrazine is attributed to the high SSA, porous structure and aromaticity of the biochars. This study shows that out of the six different feedstocks, CH- and P-derived biochars performed best, as reflected by the comparatively higher adsorption capacity and low desorption percentage, respectively. The results from this study demonstrate that the use of invasive plant species (AP and BP) and agricultural residues (CH and P) to produce biochars and their application will provide a cost-effective and eco-friendly approach to deal with pesticide contamination in the environment.

## 4. Conclusions

Atrazine is a cost-effective herbicide and is widely used for agricultural crops in the US. However, due to the mobile nature of atrazine, it can easily be lost from the soil profile to subsequently cause groundwater contamination and human health issues. Most atrazine sorption studies in Florida were conducted in calcareous or carbonatic soils. Research on atrazine sorption kinetics in an organic-rich soil is limited. This research evaluated the effects of biochars produced from six different feedstocks and their comparative ability to retain atrazine in an organic-rich soil. We hypothesized that the behavior of atrazine in organic-rich soil would be different than in a mineral soil. The outcomes from this study will also enrich the knowledge base on atrazine sorption with or without biochar applications in a soil where OM can play a significant role in sorption mechanisms. Agricultural waste materials are difficult to manage and we found that biochars made from coconut husk (a waste product of coconut) performed best in controlling the sorption behavior of atrazine in soil. Other biochars from native plant species (loblolly pine and pecan shell) also performed well and were about 7–10% better in increasing atrazine adsorption compared to biochars from invasive plant species (Australian pine and Brazilian pepper). A large quantity of coconut husk is produced in South Florida, therefore, turning that waste material into a useful resource (biochar) for soil amendment and crop production will increase overall agricultural sustainability. The result also indicates that climatic conditions may have a greater effect on the performance of the biochars in agricultural soil. In some cases, unamended soils performed (adsorption–desorption kinetics) similarly to soils amended with biochars made from Australian pine and Brazilian pepper. However, it should be noted that the unamended soil was organically rich (>15% SOM content), so, some adsorption–desorption behavior in unamended soil was not unexpected. Pyrolysis

temperature (350 °C vs. 500 °C) had effects on physicochemical properties of the biochars produced, however, no significant effect of temperature was found for atrazine adsorption. Cation exchange capacity and specific surface area were the two major properties of the biochars that contributed towards their potentiality in atrazine sorption isotherms. Overall, our experiments indicate that feedstock types and agroclimatic conditions had greater effects on biochar performance than the production temperature of those biochars.

**Author Contributions:** Conceptualization: S.G. and K.J.; Methodology: S.G. and K.J.; Formal Analysis: S.G., D.W.W. and A.R.B.; Investigation and Data Curation: S.G. and S.C.; Resources: S.G., A.R.B., J.M.N., D.W.W., C.W. and K.J.; Writing—Original Draft Preparation: S.G.; Visualization: S.G., S.D., S.C. and K.J.; Writing—Review and Editing: S.G., S.D., S.C., A.R.B., J.M.N., D.W.W., C.W. and K.J.; Supervision: K.J.; Project Administration: S.G. and K.J.; Funding Acquisition: K.J. All authors have read and agreed to the published version of the manuscript.

**Funding:** This research was funded by USDA-NIFA-Hispanic Serving Institutions Education Grant 2015-38422-24075.

**Institutional Review Board Statement:** Not Applicable.

**Informed Consent Statement:** Not Applicable.

**Data Availability Statement:** All data is available in the manuscript.

**Acknowledgments:** The authors would like to thank Herma Pierre (USDA-APHIS-PPQ S&T, Miami, FL, USA) and Mark D. Kershaw (Florida International University) for their assistance with the high-performance liquid chromatography (HPLC) system and Rosemary Hickey-Vargas and Andrew Joseph (Florida International University) for assisting with the inductively coupled plasma–optical emission spectroscopy (ICP–OES) instrument.

**Conflicts of Interest:** The authors declare no conflict of interest.

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
