# Peer review of "Physiochemical Characterization of Biochars from Six Feedstocks and Their Effects on the Sorption of Atrazine in an Organic Soil"

_agronomy, doi:10.3390/agronomy11040716_

Round 1

Reviewer 1 Report

The corrections suggested for the English language and comments lines 100-104 are in the attached file

The novelty of the results should be better highlighted in the conclusions.

Only the statistics program used is mentioned. The method must also be mentioned.

Table 2: adjust the table margins or split into two tables

Refence: check line 806

Author Response

Reviewer # 1:

Our response: Thank you for your time and effort in reviewing our manuscript, and providing valuable suggestions. We incorporated all the suggestions and the changes were highlighted in this revised manuscript.

Comment # 1: The corrections suggested for the English language and comments lines 100-104 are in the attached file

Our response: Changes were incorporated and highlighted in the revised manuscript. Please see lines 100-104 for changes.

Comment # 2: The novelty of the results should be better highlighted in the conclusions.

Our response: The results section has been updated. Please see lines 610-618, 623-625 for changes.

Comment # 3: Only the statistics program used is mentioned. The method must also be mentioned.

Our response: Detailed information of statistical method has been incorporated in the manuscript. Please see lines 267-275 for changes.

Comment # 4: Table 2: adjust the table margins or split into two tables

Our response: Table 2 has been adjusted and it has been rotated to fit the page. Please see line 178 for changes.

Comment # 5: Reference: check line 806

Our response: Thank you for bringing this to our attention. However, the citation has been removed from the manuscript to reduce the volume of the reference list. Please see lines 857-858 for changes.  

Reviewer 2 Report

The topic of the submitted manuscript is not new, but the results are original and worth of consideration. The experiment was properly projected and implemented, the applied analytical and statistical methods are suitable. Manuscript meets the scopes of the journal and probably will focus international concern. However, some minor shortcomings should be resolved before the final editor's decision.

  1. First, the manuscript is too long that makes the presentation boring and hard to read in some sections. In particular, the discussion of some basic knowledge on the biochar properties or the relationships between conditions of production and biochar properties seems not necessary. Is the justification of very old standard really necessary (like in lines 136-138)? Are the introductions to tables necessary (like in line 166)?
  2. The list of cited papers is too long and outdated. Only 17% of papers were issued after 2015, whereas really lot of new contributions was issued recently! This may influence the conclusion in line 124.
  3.  It is unclear, why some methods are extensively described in the paragraph text, while some are only briefly listed in the table captions.
  4. Could you explain the difference between C and OC in table 1. Explain the sense of OC calculation if C was measured directly.
  5. The PCA graph was extensively described (lines 368-381), but still the sense of PC1 and PC2 has not been explained. Please consider factor influencing the biochar properties, not the properties alone (as PCs)!
  6. Explain the abbreviations in the caption to table 1 (biochars) and give only the reference to this explantion under the further tables.
  7. The readability of teh figures 1, 2 and 4 is poor due to little fonts. Maybe turning the fig. 2 to landscape view give a chance for larger symbols of biochars.

Author Response

Reviewer # 2:

The topic of the submitted manuscript is not new, but the results are original and worth of consideration. The experiment was properly projected and implemented, the applied analytical and statistical methods are suitable. Manuscript meets the scopes of the journal and probably will focus international concern. However, some minor shortcomings should be resolved before the final editor's decision.

Our response: Thank you for your time and effort in reviewing our manuscript, and providing valuable suggestions. We incorporated all the suggestions and the changes were highlighted in this revised manuscript.

Comment # 1: First, the manuscript is too long that makes the presentation boring and hard to read in some sections. In particular, the discussion of some basic knowledge on the biochar properties or the relationships between conditions of production and biochar properties seems not necessary. Is the justification of very old standard really necessary (like in lines 136-138)? Are the introductions to tables necessary (like in line 166)?

Our response: We agree with the reviewer. We removed few paragraphs and sentences to reduce the volume of the manuscript. Track changes have been used to highlight the sentences that were removed.

Comment # 2: The list of cited papers is too long and outdated. Only 17% of papers were issued after 2015, whereas really lot of new contributions was issued recently! This may influence the conclusion in line 124.

Our response: We added new references and removed some old references. Current reference list is now shorter than the previous version. The statement made in Line #124 was made after we searched research studies available in this area and we found that systematic studies on physio-chemical characterization of biochars made from native and invasive plant species is limited.  

Comment # 3: It is unclear, why some methods are extensively described in the paragraph text, while some are only briefly listed in the table captions.

Our response: The description of methods in the manuscript has been modified where appropriate and has been highlighted using track changes. We wanted to describe some methods extensively for the advantage of the readers. Please see lines 140-142, 257-259 for changes.

Comment # 4: Could you explain the difference between C and OC in table 1. Explain the sense of OC calculation if C was measured directly.

Our response: C represents total carbon which was measured by combustion method using a CN analyzer, OC represents the organic carbon which was calculated from the organic matter content (OM) of the soil. OM was measured by loss on ignition method and OC was calculated using the formula,

OM (%) = OC (%) x 1.72

Comment # 5: The PCA graph was extensively described (lines 368-381), but still the sense of PC1 and PC2 has not been explained. Please consider factor influencing the biochar properties, not the properties alone (as PCs)!

Our response: The description of the PCA graph has been modified and shortened. Please see line 384, 388-398 for changes.   

Comment # 6: Explain the abbreviations in the caption to table 1 (biochars) and give only the reference to this explanation under the further tables.

Our response: To our understanding the reviewer is referring to table 2 with the abbreviations, changes have been made to other tables and figures accordingly. Please see lines 182 and 422 for changes.

Comment # 7: The readability of the figures 1, 2 and 4 is poor due to little fonts. Maybe turning the fig. 2 to landscape view give a chance for larger symbols of biochars.

Our response: Figure 2 has been changed to landscape view and other figures have been enlarged in size to give a better outlook. Please see lines 415, 420 and 431 for changes.

Reviewer 3 Report

The MS is adequate and relevant for the wide audience of the journal. The quality is high, the language is well-edited and the research work topics are in good reflection with the objectives of the journal.

The introduction chapter is well documented. The hypothesis and the objective are clearly defined.

A weak point is maybe the average age of the references list. There are just a few references from the last five years. Those older than 10 years should be refresh.

Reference suggestions:

Kocsis, T., Kotroczó, Z., Kardos, L., & Biró, B. (2020). Optimization of increasing biochar doses with soil–plant–microbial functioning and nutrient uptake of maize. Environmental Technology & Innovation, 20, 101191.

Verheijen, F. G., Zhuravel, A., Silva, F. C., Amaro, A., Ben-Hur, M., & Keizer, J. J. (2019). The influence of biochar particle size and concentration on bulk density and maximum water holding capacity of sandy vs sandy loam soil in a column experiment. Geoderma, 347, 194-202.

Kocsis, T., Biró, B., Ulmer, Á., Szántó, M., & Kotroczó, Z. (2018). Time-lapse effect of ancient plant coal biochar on some soil agrochemical parameters and soil characteristics. Environmental Science and Pollution Research, 25(2), 990-999.

The methods are proper as well, except for the description of the statistical data analysis. This chapter must be extended. It is not enough to describe which software was used. Please explain which test was applied for which data.

“Line 160”: Why did the authors choose exactly 350°C and 500°C temperatures for the pyrolyzation? - 500°C is still low in the case of pyrolyzation.

“Line 265”: Please clarify the "organic garden" expression. There is no description in the before about it.

The results are well evaluated. Many measurement data were generated.

Rework should be considered in the chapter "3.3 Environmental Implications". In this part, the authors present mainly literature data. The real own result lasts from line 571 to 576.

I accept the conclusions drawn from the results. Really nice work. I wish you even more success!

Author Response

Reviewer # 3:

Comment # 1 The MS is adequate and relevant for the wide audience of the journal. The quality is high, the language is well-edited and the research work topics are in good reflection with the objectives of the journal.

The introduction chapter is well documented. The hypothesis and the objective are clearly defined.

Our response: Thank you for your time and effort in reviewing our manuscript, and providing valuable suggestions. We incorporated all the suggestions and the changes were highlighted in this revised manuscript.

Comment # 2:  A weak point is maybe the average age of the references list. There are just a few references from the last five years. Those older than 10 years should be refresh.

Reference suggestions:

Kocsis, T., Kotroczó, Z., Kardos, L., & Biró, B. (2020). Optimization of increasing biochar doses with soil–plant–microbial functioning and nutrient uptake of maize. Environmental Technology & Innovation, 20, 101191.

Verheijen, F. G., Zhuravel, A., Silva, F. C., Amaro, A., Ben-Hur, M., & Keizer, J. J. (2019). The influence of biochar particle size and concentration on bulk density and maximum water holding capacity of sandy vs sandy loam soil in a column experiment. Geoderma, 347, 194-202.

Kocsis, T., Biró, B., Ulmer, Á., Szántó, M., & Kotroczó, Z. (2018). Time-lapse effect of ancient plant coal biochar on some soil agrochemical parameters and soil characteristics. Environmental Science and Pollution Research, 25(2), 990-999.

Our response: The reference list has been modified and shortened. We added the references suggested by the reviewer. Please see line 322, 785-790 and the track changes for updated reference list.

Comment # 3: The methods are proper as well, except for the description of the statistical data analysis. This chapter must be extended. It is not enough to describe which software was used. Please explain which test was applied for which data.

Our response: Detailed information of statistical method has been incorporated in the manuscript. Please see lines 267-275 for changes.

Comment # 4: “Line 160”: Why did the authors choose exactly 350°C and 500°C temperatures for the pyrolyzation? - 500°C is still low in the case of pyrolyzation.

Our response: The biochars produced in this study were also intended to use for another project involving the effects of these biochars on plant growth. Therefore, 350°C and 500°C temperatures for the pyrolyzation.

Comment # 5: “Line 265”: Please clarify the "organic garden" expression. There is no description in the before about it.

Our response:  Information on the organic garden has been incorporated under the 2. Materials and Methods - 2.1. Soil collection and analysis’ section. Please see lines 135-137.   

Comment # 6: The results are well evaluated. Many measurement data were generated.

Our response: We thank the reviewer for this comment.

Comment # 7: Rework should be considered in the chapter "3.3 Environmental Implications". In this part, the authors present mainly literature data. The real own result lasts from line 571 to 576.

Our response: "3.3 Environmental Implications" has been rewritten. Please see lines 569-581, 584-586, 591-601 for changes.

Comment # 8: I accept the conclusions drawn from the results. Really nice work. I wish you even more success!

Our response: We highly appreciate the reviewer’s comment.